# Repetitive mild traumatic brain injury affects inflammation and excitotoxic mRNA expression at acute and chronic time-points

**Matthew I. Hiskens** [1,2]*, **Anthony G. Schneiders**[1], **Rebecca K. Vella**[1], **Andrew S. Fenning**[1]

**1** School of Health, Medical and Applied Sciences, Central Queensland University, Rockhampton, Queensland, Australia, **2** Mackay Institute of Research and Innovation, Mackay Hospital and Health Service, Mackay, Queensland, Australia

* m.hiskens@cqu.edu.au

**Data Availability Statement:** The data is uploaded in the Mendeley Data Repository and can be accessed from [https://data.mendeley.com/datasets/y2rkvdjggm/1].

## Abstract

The cumulative effect of mild traumatic brain injuries (mTBI) can result in chronic neurological damage, however the molecular mechanisms underpinning this detriment require further investigation. A closed head weight drop model that replicates the biomechanics and head acceleration forces of human mTBI was used to provide an exploration of the acute and chronic outcomes following single and repeated impacts. Adult male C57BL/6J mice were randomly assigned into one of four impact groups (control; one, five and 15 impacts) which were delivered over 23 days. Outcomes were assessed 48 hours and 3 months following the final mTBI. Hippocampal spatial learning and memory assessment revealed impaired performance in the 15-impact group compared with control in the acute phase that persisted at chronic measurement. mRNA analyses were performed on brain tissue samples of the cortex and hippocampus using quantitative RT-PCR. Eight genes were assessed, namely MAPT, GFAP, AIF1, GRIA1, CCL11, TARDBP, TNF, and NEFL, with expression changes observed based on location and follow-up duration. The cortex and hippocampus showed vulnerability to insult, displaying upregulation of key excitotoxicity and inflammation genes. Serum samples showed no difference between groups for proteins phosphorylated tau and GFAP. These data suggest that the cumulative effect of the impacts was sufficient to induce mTBI pathophysiology and clinical features. The genes investigated in this study provide opportunity for further investigation of mTBI-related neuropathology and may provide targets in the development of therapies that help mitigate the effects of mTBI.

## Introduction

Mild traumatic brain injuries (mTBI) are the most common form of closed head injury [1] and may be asymptomatic or result in concussion [2]. Symptoms generally resolve spontaneously within a couple of days, however some patients report persistent cognitive dysfunction [3]. An emerging concept in mTBI research is the role of repetitive subconcussive impacts, rather than frank concussions, in driving neurodegeneration [4]. A subset of individuals who

**Funding:** This project was supported by the Australian Government Research Training Scheme. Funding to publish this study was provided by the Mackay Hospital and Health Service and the Mackay Institute of Research and Innovation. The funders had no role in study design, data collection and analysis, decision to publish, or preparation of the manuscript.

**Competing interests:** The authors have declared that no competing interests exist.

sustain repetitive subconcussive mTBI develop chronic consequences of these injuries including decline in cognitive function, dementia, and neurodegenerative diseases [5]. However, the aetiology of chronic neurodegeneration stemming from repetitive mTBI is poorly understood [6].

There are concurrent and self-exacerbating processes that are triggered in response to mTBI. Physical and chemical damage lead to synaptic influx and inhibited reuptake of neurotransmitters that leads to calcium dysregulation in the process known as excitotoxicity [7]. This results in breakdown of postsynaptic structure and axonal damage, and compromised transport of energy and organelles within the cell [8]. In response to these processes, inflammatory mechanisms are initiated by the microglia in order to repair damage, however this defence may be overwhelmed and serve to exacerbate the excitotoxic response [9]. The concept of the 'window of cerebral vulnerability' has been hypothesised in explaining the exacerbation of negative outcomes when repeated impacts are sustained in a short period of time [10]. There remain many questions regarding these concepts and the possible progression to chronic detriment, such as the number and severity of impacts, the duration between impacts, and the pathological mechanisms driving neurodegeneration.

Animal models are a common method for investigating the outcomes of head trauma [11]. To investigate the pathology induced by mTBI, a key requirement is a model that incorporates forces on the brain that are clinically relevant to human injury, including both linear and rotational acceleration and deceleration forces, causing diffuse injury [12, 13]. Recently, several models have been developed that utilise these biomechanics, and in doing so have moved away from models inducing focal damage indicative of moderate or severe TBI [14]. These recent studies have used a varied number of impacts between 1 and 42 [15, 16], and have focused on measuring cognitive outcomes [17], glial cell activation, neuronal damage, and aggregation of proteins such as phosphorylated tau [18]. Despite this work, pathways underpinning these processes require additional examination.

This study aimed to examine the cognitive, biochemical, and molecular changes resulting from repetitive mTBI in mice at very low impact thresholds. Three different impact totals were used to assess the possibility of a dose-dependent relationship with pathology and cognition. Behavioural changes were investigated with the use of tests previously demonstrated to evaluate acute and chronic mTBI symptoms involving neurological function and spatial learning and memory. mRNA expression changes in hippocampus and cerebral cortex were examined for neuronal damage with tau protein (MAPT), TDP-43 (TARDBP), and neurofilament light (NEFL); glial response with glial fibrillary acidic protein (GFAP) and allograft inflammatory factor 1(AIF1); excitotoxicity with glutamate ionotropic receptor AMPA type subunit 1 (GRIA1); and inflammation with C-C motif chemokine ligand 11 (CCL11) and tumour necrosis factor (TNF) (see discussion for a detailed description of gene selection and S1 Fig for a module map of the connection between these genes). Serum changes in levels of Tau and GFAP were assessed to investigate biochemical changes. It was our hypothesis that increasing numbers of mTBIs would have a cumulative effect on chronic behavioural deficits, levels of protein damage in collected sera, and mRNA expression of axonal damage, astrocyte reactivity, neuroinflammation and excitotoxicity genes in this murine model of repeated mTBI.

## Materials and methods

### Animals and general overview

Experimental procedures were approved by the Animal Ethics Committee of Central Queensland University (CQU AEC 0000020614) under guidelines from the National Medical Research Council of Australia. The ARRIVE guidelines were adhered to for the design and

reporting of the study. A total of 64 male C57BL/6J mice (Animal Resource Centre, Canning Vale, WA, Australia) were housed in a constant 12:12 hour light-darkness cycle, with the temperature controlled at 22 ± 2˚C. Mice were housed four to six per cage, and food and water access as permitted *ad libitum*. At the time of arrival mice were 8 weeks old and undertook a two-week habituation period to allow acclimatisation to their new environment before the initiation of the study protocol. Bodyweight of each mouse was assessed before the commencement of mTBI administration, weekly during administration, and at time of euthanasia.

## Groups

There were two separate study arms: an acute branch of animals sacrificed 48 hours following final impact, and a chronic branch of animals were sacrificed 90 days following final impact (Fig 1). Mice were randomised using the random number generator function of Excel to one of four protocols: i) a single impact (1-IMP); ii) five total impacts (5-IMP); iii) 15 total impacts (15-IMP); and iv) control (CON). For all groups not receiving 15 impacts, sham procedures were undertaken on corresponding impact days whereby mice were anesthetised on the same schedule as the 15-impact group, but no injuries were administered. In this way, CON received 15 sham anaesthesia bouts, 1-IMP received 14 sham bouts, and 5-IMP received 10 sham bouts, to control for the possibility of anaesthesia interacting with injury and influencing function. Impacts or anaesthesia for all groups were delivered across a 23-day span, on a rotation of three impact/sham days followed by 2 rest days (Fig 2A). Following final injury, groups were assessed in behavioural measures and samples were collected (Fig 2B). No animal in any group died during impact or in the recovery phase, there was no evidence of bleeding or skull fracture at post-mortem analysis of any animal, and no animals were excluded from analysis.

## mTBI modelling

To mimic the head acceleration forces sustained in human mTBI, mice were subjected to mTBI via an apparatus designed and built for this purpose, as previously described [19]. An enclosed inhalation chamber (1 L) containing 0.5mL of isoflurane (Zoetis, Rhodes, NSW, Australia) in a cotton ball (yielding a steady 4% concentration) was used to anesthetise the mice. Inhalation between 1 to 2 minutes resulted in light anaesthesia, as determined by lack of response to tail pinch. A steel weight (12mm diameter) of 25g was used for impacting the skull. In attempting to model subconcussive impact, this 25g weight is a considerable development, as the lowest weight previously used in an animal model was 53g by Mannix and colleagues [14]. The weight was dropped from a height of 1m and guided through a PVC tube (15mm diameter). A small rubber cap (1x10mm) was attached to the bottom of the weight to restrict the contact zone. Prior to mTBI, the mouse was positioned chest down on the apparatus platform, which consisted of two magnetically adjoined acetate panels. This platform could

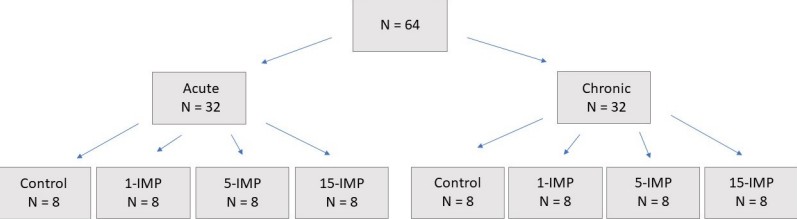

**Fig 1. Group allocations for the acute and chronic arms of each treatment.** 64 total mice were used for the study, with N = 8 randomly allocated to each group.

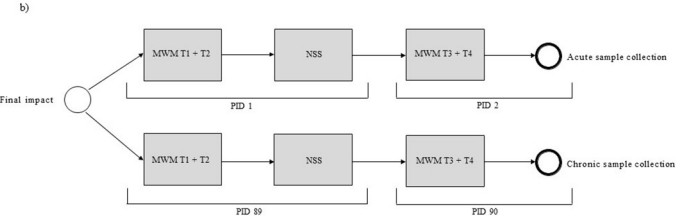

**Fig 2. Schedule of involvement in treatment and behavioural assessment.** (a) All mice were involved in either sham anaesthesia or impact conditions on 15 out of 23 days. 'A' = anaesthetic only; 'I' = impact. (b) Behavioural testing involved MWM trial 1 + 2 and NSS on PID 1 for acute groups and PID 89 for chronic groups, followed by MWM trial 3 + 4 and sample collection on PID 2 for acute groups and PID 90 for chronic groups.

support a maximum weight of 33g, so that the platform collapsed upon impact, resulting in minimal platform resistance applied to the head of the mouse. The head was positioned under the vertical tube, through which the impact weight protruded. Specific alignment was such that the weight made contact between the bregma and lambda intersection. Upon impact, the mouse fell and rotated about a horizontal axis, and landed in a supine position on the padded landing area 10 cm below the stage, which was composed of a sponge cushion (15cm length x 9cm width x 7.5cm depth). The impact weight was tethered to the guide tube by commercially available braided nylon line (Spear and Jackson, Melbourne, VIC, Australia), restricting the fall of the weight so that it could continue downward no more than 1 cm beyond the starting position of the dorsal surface of the skull, thereby avoiding unintentional secondary contact. Following impact, the mouse was immediately moved to a recovery heating pad, and recovery was monitored.

## Neurological restoration

To assess neurological restoration following mTBI, time to recover righting reflex (RR) was monitored following isoflurane-induced anaesthesia (controls) or mTBI (n = 8 per group). Mice were placed in a supine position on the recovery pad, and the time taken for the animal to adopt a prone position was recorded after each impact or anaesthetic administration. RR time was calculated from the discontinuation of isoflurane inhalation to the first sign of righting.

## Neurological impairment

Neurological impairment of mTBI mice compared with controls was assessed via neurological severity score (NSS), which is a composite measure of motor function, alertness and behaviour in rodent models of TBI [20]. NSS consists of a series of ten tests that are undertaken on a pass/fail basis and is a reliable predictor of outcomes [21]. One point is scored for inability to complete each of these actions, with a maximal score of ten indicating a failure of all tasks and severe neurological dysfunction. NSS was assessed at post-injury day 1 (PID 1) for the acute

mTBI groups, and at PID 89 for the chronic groups (n = 4 per group). The researcher assessing NSS was blinded to the condition of the animal.

## Morris water maze

Control and mTBI groups were tested in the Morris water maze (MWM), which provides a measure of hippocampal-dependent spatial learning and memory [22]. The test was conducted in a circular tank with diameter of 110cm, with highly visual cues fixed at locations around the pool. The pool was filled with water (temperature 27 +/- 1˚C) made opaque with nontoxic, water-soluble Tempera paint (Fine Art Supplies, Auckland, NZ). A round platform with a diameter of 10cm was hidden 1cm below the surface of the water in the northern quadrant. A total of four trials were administered across two consecutive days, with a 6-hour interval was provided between trials on the same day, as described previously [23]. Each trial consisted of 3 attempts to reach the hidden platform, with a start position from each of the quadrants that did not contain the platform (south, east and west). For each trial, a random order of start positions was selected, and this was held consistent for each animal across the trial. The trials occurred on PID 1 and 2 for the acute groups, and PID 89 and 90 for chronic animals (n = 4 per group). For each attempt, mice were given a maximum test duration of 60 sec to find and remain on the hidden platform. Mice that did not locate the platform within the allocated time were guided to the platform and allowed to rest for 10 sec. On PID 2 and 90, all animals also underwent a probe trial, which involved the hidden platform being removed from the pool. Mice were placed in the pool opposite the target quadrant (quadrant where the platform had been) and had a time limit of 30 sec to search for the platform. Time spent in the target quadrant was assessed, as described previously [22]. The researcher assessing MWM was blinded to the condition of the animal. Animals were assessed for motor function deficits using Kinovea 0.8.15 software to track swim speed, and time spent in the goal quadrant during the probe trial.

## Sample collection

On PID 2 and 90, euthanasia was administered via inhalation of isoflurane between 4 to 6 minutes, with death confirmed by cessation of breathing. Blood samples were collected via the inferior vena cava and allowed to clot before being centrifuged at 5000 rpm for 15 minutes. Sera was aliquoted and stored at -80˚C. The brain was removed and weighed, then washed in ice cold oxygenated (95% $O_2$, 5% $CO_2$) artificial cerebrospinal fluid (CSF) containing 118.0 mM NaCl, 3.5 mM KCl, 1.3 mM $MgCl_2$, 26.2 mM $NaHCO_3$, 1.0 mM $NaH_2PO_4$, 2.5 mM $CaCl_2$, 11.0 mM glucose, before being rapidly dissected on a frozen dissection platform for hippocampus and cerebral cortex sections. Sections were frozen at -80˚C for genetic and biochemical analysis. To enable blinding conditions, collection tubes were coded so that group names were not accessible to the investigators undertaking sample analysis. Coding information was secured on the lead investigators computer, and the codes were only accessed after the samples were analysed.

## Quantitative real-time reverse transcriptase PCR

mRNA was extracted from tissue homogenates of the hippocampus and cerebral cortex of mTBI and sham-injured mice (n = 4 per group) using the phenol-chloroform method [24]. Sample concentration and purity were evaluated using a spectrophotometer (NanoDrop 2000c, Thermo Fisher Scientific, Wilmington, DE, USA) (mean ± SD 260/230 spectral ratio: 1.92 ± 0.22; mean ± SD 260/280 spectral ratio: 2.00 ± 0.05). Complementary DNA was synthesised using Superscript III First-Strand Synthesis System for reverse transcriptase-PCR

according to the manufacturer's instructions (Applied Biosystems, Foster City, CA, USA) and run in a thermal cycler (T100 Thermal Cycler, Bio-Rad, Gladesville, NSW, Australia). Five genes commonly used as indicators of the presence and severity of head trauma were analysed from the cortex and hippocampus tissue; MAPT, GFAP, AIF1, TNF, and NEFL. In addition to these, three genes rarely investigated in mTBI conditions were compared across injury groups; GRIA1 CCL11, and TARDBP. Samples and negative controls were prepared in duplicate using Taqman universal PCR master mix and run using a thermal cycler (Rotor-Gene Q, Qiagen, Venlo, Netherlands). The following Taqman gene expression assays were used (Applied Biosystems catalogue numbers): mouse MAPT (Mm00521990_m1), GFAP (Mm01253030_m1), AIF1 (Mm00479862_g1), GRIA1 (Mm00433753_m1), CCL11 (Mm00441238_m1), TARDBP (Mm01257504_g1), TNF (Mm00443258_m1), NEFL (Mm01315666_m1), and gene products were normalized to endogenous mouse GAPDH (Mm99999915_g1). Relative expression for Taqman-analysed transcripts was calculated using the delta-delta Ct method [25].

## Biochemical assessment via Enzyme-Linked Immunosorbent Assay (ELISA)

Serum tau phosphorylated at threonine 231 (p-tau 231) and GFAP protein levels were qualified by ELISA kits (p-tau MBS9356404, GFAP MBS2089651) following the manufacturer's instructions (MyBiosource, San Diego, CA, USA). All standards, positive and negative controls, and samples were run in duplicate. 96-well immunoplates (Corning Costar, Corning, NY, USA) were coated with 100 μl of capture antibody and incubated overnight at 4˚C. Nonspecific binding was blocked with blocking buffer. 100 μl of samples and standards were then added to the coated wells for 1 hr at room temperature. After incubation, 100 μl of the working biotinylated detection antibody was added to each well and incubated for a further 1 hr. 100 μl of streptavidin-HRP was added to each well and incubated for 30 mins at room temperature. 3,3′,5,5′-tetramethylbenzidine was added to start the colour reaction. The reaction was stopped after 10 min with 1 M HCl solution, and the absorbency was immediately measured at 450 nm (iMark plate reader, Bio-Rad, Gladesville, NSW, Australia). All samples fell within normal range of the standard curve, which was 6.25pg/ml to 200pg/ml for p-tau and 62.5 to 4000 pg/ml for GFAP.

## Statistical analysis

Group numbers for behavioural and laboratory tests were calculated via an a priori power analysis using α of 0.05, power of 0.8, and means and SD from previously laboratory pilot data. Statistical analysis were performed using IBM SPSS Statistics for Windows Version 25.0 (IBM Corp, Armonk, NY). Data were evaluated for normality (Shapiro-Wilks test or Kolomogov-Smirnov test) prior to statistical testing. As all data was parametric, 1-way ANOVA, or repeated-measures 2-way ANOVA, with Tukey post-hoc tests (alpha <0.05) was used to assess for statistically significant differences. All data are presented as means with standard deviations.

## Results

Mice subjected to mTBIs in the 1-IMP, 5-IMP and 15-IMP groups showed no signs of convulsions or physical stress following impacts, indicating that our model sufficiently mimicked the mild impact forces typically seen in sub-concussive injury. Due to the asymptomatic nature of the injuries, no mice were withdrawn from the study on ethical grounds.

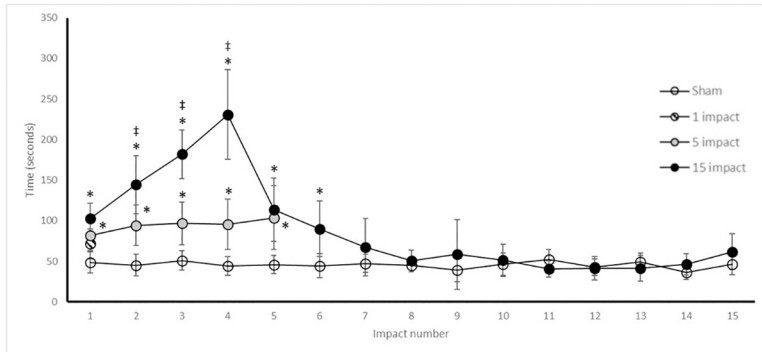

**Fig 3. Repetitive mTBI results in transient delay in recovery of righting reflex.** Time to regain righting reflex (seconds) following impact or sham control anaesthesia. Values reported as mean (± SD). $^*$ $p < 0.05$ difference compared with sham control; ‡ $p < 0.05$ difference with 1-IMP; # p $< 0.05$ difference compared with 5-IMP. N = 8 per group.

## Righting reflex

Impact caused a significant increase in the time required to regain consciousness (Fig 3). Compared with controls, all impact groups took significantly longer to recover the righting reflex after one impact ($p < 0.05$). RR latency in the 5-IMP group was significantly increased compared to control throughout the entire impact schedule ($p < 0.05$). The increased latency to recover the righting reflex persisted in the 15-IMP group for impacts 1–6, compared to control ($p < 0.05$). For impacts 7 through 15, recovery times were not significantly different in the 15-IMP group from controls ($p < 0.05$). The 15-IMP group RR time was significantly greater than 5-IMP group following impacts 2 through 4, but not for impacts 1 and 5 ($p < 0.05$). The 1-IMP RR time was not significantly different to control.

## Spatial learning and memory

Hippocampus-dependent spatial learning and memory was assessed at acute (Fig 4A) and chronic (Fig 4B) time points using the MWM. There were no differences in swim speed

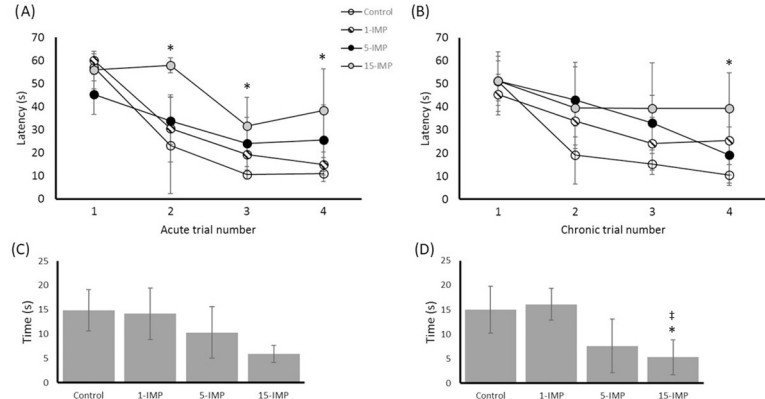

**Fig 4. Repetitive mTBI impairs performance in the Morris water maze.** (A) Time to find the hidden platform (seconds) in the MWM at acute testing. (B) Time to find the hidden platform in the MWM at chronic testing. (C) Time spent in the goal quadrant of the Probe Test at acute testing. (D) Time spent in the goal quadrant of the Probe Test at chronic testing. Values reported as mean (± SD). $^*$ $p < 0.05$ difference compared with sham control; ‡ $p < 0.05$ difference with 1-IMP; # p $< 0.05$ difference compared with 5-IMP. N = 4 per group.

between injured and control mice (0.26 +/- 0.01m/s for 1-IMP; 0.26 +/- 0.01m/s for 5-IMP; 0.28 +/- 0.01m/s for 15-IMP; 0.28 +/- 0.01m/s for CON, *p = 0.2*), and none of the mice were excluded from MWM testing based on lack of motor function. Mice from the control and impact groups showed progressive improvement in the ability to locate the hidden platform with each subsequent test. For acutely tested mice, mTBI groups displayed latency in this ability, and post-hoc analyses found that the 15-IMP group was statistically different to control at trial 2, 3, and 4 *(p <0.05)*. There was no significant difference between 15-IMP and control at trial 1, and no differences between 15-IMP and other impact groups at any trial. No differences were seen from control with the 1-IMP and 5-IMP groups at any trial. At the chronic testing time-point, again only the 15-IMP group displayed significantly impaired ability to find the platform at trial 4 *(p <0.05)*. There were no differences with the 15-IMP group compared to control at trial 1 through 3, and no differences between 15-IMP and the other impact groups. The 1-IMP and 5-IMP group times to find the platform were not different from control at any trial.

For the probe trial testing, analyses comparing the mTBI and control groups found no significant differences in the time that mice spent searching from the platform in the goal quadrants at the acute testing time point (Fig 4C). In contrast, three months following final impact the 15-IMP group showed learning impairment as evidenced by reduced preference for the target quadrant compared to control *(p <0.05)* and compared to 1-IMP *(p <0.05)* (Fig 4D).

## Behaviour and motor function

In NSS (Fig 5A and 5B) testing at PID 1, the 15-IMP group score was significantly higher than control and both other impact groups *(p <0.05)*. For chronic testing at PID 89, all mTBI groups revealed no significant differences in score for any group compared to sham control *(p >0.05)*.

## Animal and brain weights

There were no significant differences in bodyweight between impacted and non-impacted groups at all time-points. Bodyweight (mean ± SD) at the time of the euthanasia for acute mice was 23.26 ± 1.28 g, and 28.67 ± 2.09 g for chronic mice. The average brain weight was 0.43 ± 0.02 g, and there were no significant differences between any of the groups for brain weight *(p >0.05)*.

## Quantitative reverse transcription polymerase chain reaction analysis

Due to the large amount of data generated from the molecular analysis, involving four groups, eight genes, two tissue types, and two time points, results information has been condensed

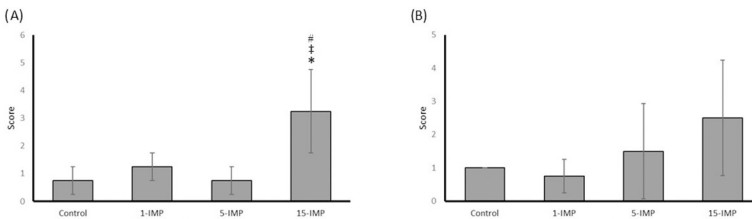

**Fig 5. Repetitive mTBI impairs acute but not chronic neurological severity score.** (A) NSS score at acute testing. (B) NSS score at chronic testing. Values reported as mean (± SD). * $p < 0.05$ difference compared with sham control; ‡ $p < 0.05$ difference with 1-IMP; # p < 0.05 difference compared with 5-IMP. N = 4 per group.

into tables that contain all F and p values for the ANOVAs with corresponding post-hoc testing where required (Tables 1 and 2). Genes were differentially expressed in cortex and hippocampus and displayed unique expression at acute time-points compared with chronic animals. For genes with altered expression following injury there appeared a dose-dependent relationship with number of impacts. In the acute cortex the 15-IMP group showed increased expression of MAPT, GFAP, and TNF genes relative to control ($p > 0.05$). In the chronic cortex AIF1, CCL11 and TARDBP levels in the 15-IMP group were elevated relative to control ($p > 0.05$). In the hippocampus, acute measurement saw the increase of GRIA1, CCL11, and NEFL in the 15-IMP group ($p > 0.05$), while chronic measurement resulted in elevated GFAP levels in the 15-IMP group and elevated AIF1 in all impact groups relative to control ($p > 0.05$). Results pertaining to cortex and hippocampal mRNA expression change for acute and chronic mice can be seen in Fig 6A–6D.

## Serum biochemical markers

For groups measured at both 48 hours and 90 days after final mTBI there were no differences between any group in serum p-tau (Fig 7A and 7B) or GFAP levels (Fig 7C and 7D, $p > 0.05$).

## Discussion

This study used a mouse model of mTBI that closely mimics the acceleration forces, impact speeds, and biomechanical properties of head impacts in humans. The impacts were delivered to the surgically unaltered head, and the parameters chosen were at the lowest impact weight reported [14]. The impacts resulted in no cases of skull fracture, macroscopic brain damage,

**Table 1. Statistical results obtained from the one-way ANOVAs for changes in expression of the eight genes of interest for the two brain regions at acute sacrifice (48-hour post-mTBI).**

| Cortex | | | | | | | | |
|---|---|---|---|---|---|---|---|---|
| | | | | | POST-HOC | | | |
| | F | P | SHAM:1-IMP | SHAM:5-IMP | SHAM:15-IMP | 1-IMP:5-IMP | 1-IMP:15-IMP | 5-IMP:15-IMP |
| MAPT | 6.40 | **.01** | .25 | .93 | **< .01** | .92 | .14 | .35 |
| GFAP | 4.88 | **.02** | 1 | .92 | **.03** | .89 | **.03** | .09 |
| AIF1 | 3.22 | .06 | - | - | - | - | - | - |
| GRIA1 | 2.40 | .12 | - | - | - | - | - | - |
| CCL11 | 1.39 | .29 | - | - | - | - | - | - |
| TARDBP | 0.72 | .56 | - | - | - | - | - | - |
| TNF | 12.22 | **< .01** | .09 | .98 | **< .01** | **.05** | .11 | **< .01** |
| NEFL | 3.37 | .06 | - | - | - | - | - | - |
| Hippocampus | | | | | | | | |
| | | | | | POST-HOC | | | |
| MAPT | 4.60 | **.02** | **.02** | .08 | .14 | .82 | .63 | .99 |
| GFAP | 1.11 | .38 | - | - | - | - | - | - |
| AIF1 | 1.57 | .25 | - | - | - | - | - | - |
| GRIA1 | 21.68 | **< .01** | .88 | .31 | **< .01** | .70 | **< .01** | **< .01** |
| CCL11 | 8.94 | **< .01** | .99 | .23 | **< .01** | .57 | **< .01** | .07 |
| TARDBP | 1.70 | .22 | - | - | - | - | - | - |
| TNF | 0.89 | .48 | - | - | - | - | - | - |
| NEFL | 5.96 | **.01** | .83 | .93 | **.01** | .99 | **.04** | **.03** |

Significant differences p < .05 are in bold.

**Table 2. Statistical results obtained from the one-way ANOVAs for changes in expression of the eight genes of interest for the two brain regions at chronic sacrifice (3 months post-mTBI).**

| Cortex | | | POST-HOC | | | | | |
|---|---|---|---|---|---|---|---|---|
| | F | P | SHAM:1-IMP | SHAM:5-IMP | SHAM:15-IMP | 1-IMP:5-IMP | 1-IMP:15-IMP | 5-IMP:15-IMP |
| MAPT | 3.51 | **.05** | .49 | .86 | .41 | .90 | **.04** | .13 |
| GFAP | 1.33 | .31 | - | - | - | - | - | - |
| AIF1 | 6.44 | **< .01** | **.03** | .18 | **< .01** | .69 | .83 | .26 |
| GRIA1 | 6.85 | **< .01** | .08 | .48 | .36 | **< .01** | .76 | **.03** |
| CCL11 | 4.50 | **.03** | .10 | .46 | **.02** | .74 | .78 | .25 |
| TARDBP | 10.66 | **< .01** | .12 | .67 | **< .01** | .57 | .06 | **< .01** |
| TNF | 1.26 | .33 | - | - | - | - | - | - |
| NEFL | 0.76 | .54 | - | - | - | - | - | - |
| Hippocampus | | | POST-HOC | | | | | |
| MAPT | 2.94 | .08 | - | - | - | - | - | - |
| GFAP | 6.40 | **< .01** | .98 | .66 | **.02** | .47 | **.01** | .19 |
| AIF1 | 12.87 | **< .01** | **.03** | **.02** | **< .01** | .96 | .07 | .22 |
| GRIA1 | 0.46 | .72 | - | - | - | - | - | - |
| CCL11 | 1.08 | .40 | - | - | - | - | - | - |
| TARDBP | 2.65 | .10 | - | - | - | - | - | - |
| TNF | 2.96 | .08 | - | - | - | - | - | - |
| NEFL | 1.71 | .22 | - | - | - | - | - | - |

Significant differences p < .05 are in bold.

subdural haematoma, or death, in keeping with human mTBI pathology. The loss of consciousness times in the impact groups were typical of those reported in other mTBI rodent studies [26]. Time under anaesthesia was minimised to reduce the effect on cognition and

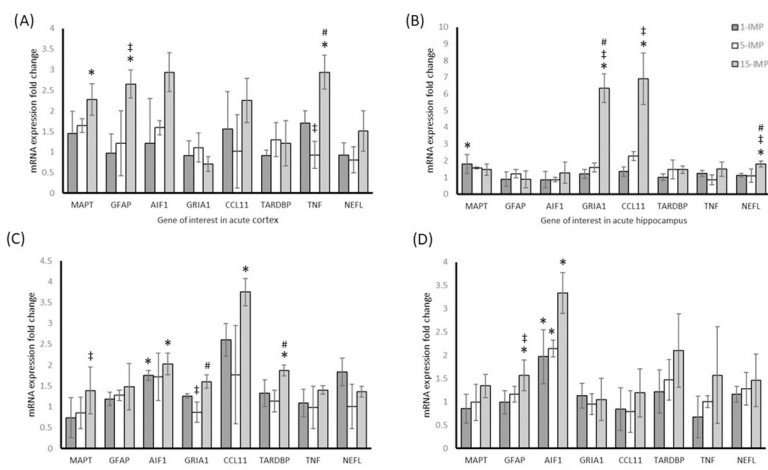

**Fig 6. Repetitive mTBI induces upregulation of neurodegenerative genes.** (A) mRNA expression fold change relative to sham control of eight genes in the cortex at acute testing. (B) mRNA expression fold change relative to sham control of eight genes in the hippocampus at acute testing. (C) mRNA expression fold change relative to sham control of eight genes in the cortex at chronic testing. (D) mRNA expression fold change relative to sham control of eight genes in the hippocampus at chronic testing. Normalised to Gapdh (± SD). * $p < 0.05$ difference compared with sham control; ‡ $p < 0.05$ difference with 1-IMP; # $p < 0.05$ difference compared with 5-IMP. N = 4 per group.

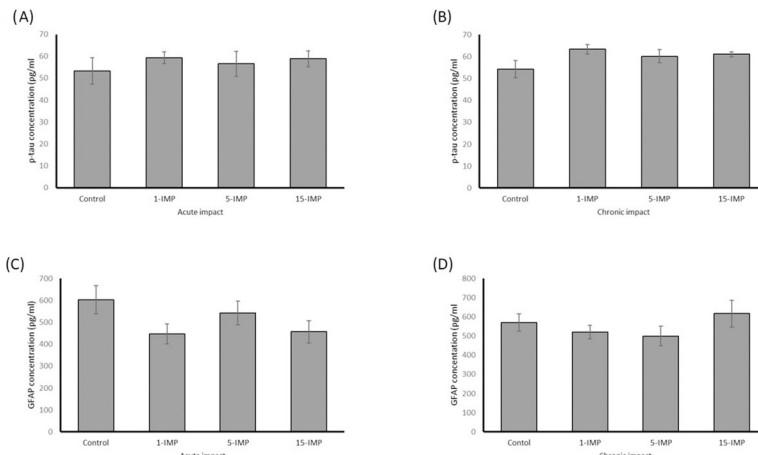

**Fig 7. Repetitive mTBI does not affect p-tau or GFAP protein expression in our model.** (A) P-tau serum protein expression at acute testing. (B) P-tau serum protein expression at chronic testing. (C) GFAP serum protein expression at acute testing. (D) GFAP serum protein expression at chronic testing. No significant differences were seen between any groups. N = 4 per group.

pathology, and sham controls received the same anaesthesia protocol. With these factors, and due to the mild deficits resulting from impact, it could be argued that this model provides impacts equivalent to human sub-concussive mTBI [27]. Repetitive sub-concussive impact in humans has been linked with chronic neurodegeneration [28], while single events may be innocuous [2]. In keeping with this concept, this study found that differing numbers of mTBI resulted in a dose-dependent response in behavioural measures and molecular signalling, whereby the 15-impact model produced the most pronounced changes in the acute phase, and resulted in CTE-like pathology when assessed chronically.

Neurological restoration, as measured by restoration of RR, was initially impaired in the 15-IMP group, as effects appeared to accumulate for the first 4 impacts. Interestingly, a sharp improvement was seen for impact 5, which persisted throughout the remainder of the impacts and resulted in neurological restoration times not different from the control group. This was despite consistent force of impact and did not coincide with the two-day 'rest period' of the impact schedule. It is unlikely that anaesthesia tolerance is a factor in this response, as the sham control receiving only anaesthesia did not exhibit a similar reduction in RR time. This phenomenon has been observed in previous studies, and these authors have hypothesised that a decrease in RR time may be as a result of CNS adaptation and initiation of neuroprotective pathways in response to repeated mTBI [29]. The exact mechanism of improvement is unable to be speculated upon in this study, as analysis was only undertaken after the full complement of 15 impacts. While the 5-IMP group had RR times significantly higher than control, this group did not see the same peak as the 15-IMP group, suggesting that having impact free days after two impacts rather than after the three impacts of the 15-IMP group served as an opportunity for healing such that accumulated detriment was not seen. At impact 5, the 5-IMP and 15-IMP groups were not significantly different, and it would be interesting to see if the 5-IMP group observed the same improvement in RR with additional impacts. The 1-IMP group displayed no significant increase in RR compared with controls, suggesting unremarkable change resulting from a single impact.

Findings from the behavioural data include differences in NSS in the acute phase of recovery 24 hours after final injury, but no differences between groups in the NSS at three months. Previous studies using a similar acceleration mTBI mechanism have shown NSS detriment at 1

hour through 7 days [30], and 24 hours [31], although others have also found no detriment [32]. The data shown here suggests that even though mild impacts in our model were administered below the threshold seen previously in similar studies, there was still measurable neurological disturbance manifested in behavioural testing. This is key when translating to human research, where an emerging concept is in the repetitive subclinical injuries that are sustained in environments such as on a football practice field. While repeated mild blows to the head may not result in measurable changes in field-side tests such as the SCAT5, pathological disturbances leading to long term clinical symptoms might still be present [33].

Repetitive mTBI groups displayed spatial learning and memory deficits in acute MWM and probe trials that persisted in chronic testing. Differences between groups became more pronounced throughout the trial schedule, with the control group achieving maximum speed in finding the platform at trial 3, and the IMP-5 and IMP-15 groups displaying ongoing learning deficits until the final trial attempt. In the probe trial, the groups that received repetitive mTBIs spent significantly less time in the target quadrant searching for the platform. This is in line with previous studies [34, 35] that have reported persistent memory impairment following repetitive mTBI, with comprehensive investigations giving insight into poor outcomes 12 months post-injury when there are short inter-injury intervals [36]. The differences are also in line with emerging clinical data which suggests that repetitive impacts to the head are implicated in subacute and chronic neurological deficits. Cognitive disruption has shown a higher prevalence in retired profession football players compared with matched healthy controls [37], and in former athletes who sustained a sports concussion more than 30 years before testing [38].

The purpose of this paper was not to identify a predictive and reliable biomarker of mTBI, but to provide additional details regarding the signalling processes within the brain following varying levels of injury. This study had two key goals: (1) to ensure that the subtle level of repetitive impact in our model was enough to induce mRNA expression changes in genes that have been associated with brain injury, and (2) that it allowed the investigation of novel genes that allow further understanding of pathways of pathology. Although there are numerous genes we could have analysed, we selected specific genes for investigation based upon the role they may play in the excitotoxicity, inflammation, and neurodegeneration in the context of mTBI. GRIA1 was assessed as a measure of excitotoxicity, which is a trigger of neurodegenerative cascades and is implicated in the disruption of spatial working memory in the hippocampus. To examine axonal neurodegeneration, MAPT and NEFL were selected, as they are indicators of structural damage and are involved in chronic neurofibrillary tangle formation and neurofilament breakdown, respectively. TARDBP was selected for its implication in these neurodegenerative processes and role in protein signalling and organelle transport within the neuron. GFAP and AIF1 were used as a measure of activated astrocytes and microglia, respectively. Glial cells are activated in response to injury where they promote neuroinflammation that is aimed at neurological recovery and repair. Finally, TNF and CCL11 were used as measures of inflammation, with TNF a classic pro-inflammatory cytokine that has been measured in TBI conditions, and increased CCL11 production found in brain aging and disease. As expected, the brain structure and the timing of analysis did influence the expression of the eight genes examined. Typically, for genes that were responsive to injury, a dose-dependent relationship was seen, whereby the highest increase in gene expression was seen in the 15-IMP group and the lowest increase in the single impact group.

Excitotoxicity is an immediate consequence of mTBI, and involves the rapid synaptic influx and inhibited reuptake of neurotransmitters and amino acids which can result from mTBI [39]. Glutamate is the primary excitatory neurotransmitter involved in this process [40, 41], and injury leads to unregulated accumulation of glutamate [42]. This over-activates

downstream signalling pathways leading to an uncontrolled surge in intracellular calcium concentration, which is an underlying mechanism of neuronal death [43, 44]. In response to the increase in glutamate in the synaptic cleft, excessive activation of N-methyl-D-aspartic acid (NMDA) receptor GluA1 (coded by the GRIA1 gene) attempts to clear these metabolites [45]. The resulting neurotoxicity involves the breakdown and loss of postsynaptic dendrites and cell bodies [8], which compromises synaptic plasticity and learning [46]. The hippocampus is more susceptible to excitotoxic injury than other parts of the brain [47], and this explains the significant increase in GRIA1 mRNA expression in the hippocampus at the acute timepoint in our study. At chronic measurement, GRIA1 was not increased in any of the impact groups compared with control, which is fitting with the acute mechanism of this response.

Microtubule associated protein-tau (MAPT) is specific to the axonal region of the neuron and is required in the organisation and construction of microtubule bundles [48]. Under conditions of neurochemical or physical trauma, MAPT is disrupted, leading to compromised transport of proteins and organelles in the axon, which ultimately leads to the formation of disruptive neurofibrillary tangles and neuronal death [49]. This study found that MAPT was elevated in the cortex in the days following injury, and this increase persisted at three months. This is highly relevant in validating this as a model representative of human mTBI, as persistent and progressive tau pathology is a defining feature of the neurodegeneration seen in chronic traumatic encephalopathy (CTE) [50]. In contrast, in the hippocampus there was an acute increase in MAPT expression that was not persistent at late chronic sampling. This is in line with the pathology seen in human CTE data, where abnormal tau aggregation and neurofibrillary tangles are seen in the sulci of the cortex but less commonly in the hippocampus [50].

Neurofilament light (NF-L) is a structural protein of myelinated axon cytoskeleton within white matter regions [51]. Neurotrauma initiates breakdown and release of neurofilament chains, which can be measured in tissue, CSF, and serum [52]. At acute measurement, NEFL mRNA expression (indicating NF-L upregulation) was significantly elevated in the hippocampus region, and while levels were increased in the cortex, this trend did not reach statistical significance. At 3 months there was no difference from controls in either the cortex or the hippocampus. In human studies, CSF and blood measures of NF-L have been shown to provide a sensitive measure of trauma at both acute and chronic time points. In American football players, impact within an hour of contact showed correlation with the number and magnitude of head impacts sustained [53], although acute measures have not always been elevated in clinical settings [54]. When monitoring long term recovery, CSF and serum levels have been correlated with outcomes 6 and 12 months after injury [51]. NF-L has also been implicated in chronic pathology and progressive neurodegeneration, where post-mortem plasma levels were correlated with cognitive impairment and severity of NFT pathology [55].

TDP-43 is involved in maintaining the expression of correct isoform ratios within the neuronal cytoskeleton [56]. It is upregulated as a result of the elongation and stretching of axons during acceleration induced brain deformation, in order to undertake repair and reorganization of cytoskeleton microtubules and neurofilaments [57]. However, in conditions of repeated trauma TDP-43 demonstrates increased dysregulation and aggregation, leading to disruption of neural signalling and tau NFT pathology [58, 59]. Indeed, in progressive cases of CTE, abnormal TDP-43 expression in the cortex has been found to increase at the same rate and clinical stages as the accumulation of phosphorylated tau [60]. In the present study, TDP-43 was assessed by TARDBP gene expression, and was significantly elevated in the cortex of the 15-IMP group at the chronic measurement. There was no evidence of early brain accumulation at the acute time-point, which follows the development seen in human cases [61]. This provides evidence of progressive pathology in this model indicative of the clinical pathology that has been described in cases of CTE.

Glial fibrillary acidic protein (GFAP) is a structural protein expressed by astrocytes [62], and is used as a reliable clinical tool in differentiating mTBI patients from controls [63]. This study identified elevated GFAP mRNA levels in the PFC at 48 hours, with levels in the hippocampus not significantly different from control. Conversely, levels at three months were elevated in the hippocampus but not the PFC. This differential mRNA expression is indicative of the high level of GFAP specificity in brain tissue following injury [64], and it is likely that the two brain regions undergo different rates of healing and recovery that reflect this difference. In human studies GFAP has been used to measure neurological damage and degeneration in the hours [65, 66] and months [67] following injury, with long term elevation correlated with impaired recovery from mTBI [68]. The present study showed a similar relationship, with the 3-month hippocampus expression of GFAP correlated with performance in the MWM, a measure of hippocampus dependent spatial learning and memory. GFAP mRNA expression was not different from control in the 1-IMP and 5-IMP groups, and likewise these groups had no impairment in the MWM and probe trial. However, mRNA expression was elevated in the 15-IMP group, which displayed corresponding performance deficits in the MWM and probe trial. This finding lends support for the role of repetitive injury in driving neurodegeneration, and in the value of GFAP as a tool in monitoring mTBI injury and recovery.

Microglia are the primary cells involved in regulating brain development and maintaining homeostasis through immune defence [69]. In these roles, microglia are involved in myelination, synaptic formation, neurogenesis of developing cells, and phagocytosis of apoptotic cells [70]. In response to brain trauma, microglia undergo morphological change and mount a potent inflammatory response designed to protect and repair the damaged cells [71]. However, if damage is too severe or the insult is ongoing, microglia will remain in a state of sustained defence which results in persistent inflammation and has been shown to result in neurodegeneration and functional deficits in preclinical [72] and clinical studies [73, 74]. In assessing mRNA AIF1 expression, the present study found evidence of this persistent microglial activation in the cortex and the hippocampus at three months following injury, which provides evidence of the neurodegenerative changes occurring in the model resulting from the inability to repair the damage from impact. There were also elevated AIF1 levels in the cortex of the 15-IMP group that was approaching statistical significance (p = 0.056) in the acute timepoint, and the amplification of AIF1 may have been affected by the timing of sampling, as it has been hypothesised that microglial signalling is secondary to astrocyte activation [75].

TNF is a cytokine that that has both pro and anti-inflammatory signalling properties, and is released from neurons, astrocytes and microglia after CNS insult [76, 77]. TNF is commonly assessed in TBI, and has been shown to be elevated in animal models and clinical studies [78]. At 48 hours post injury, mRNA expression was elevated in the PFC, however in the hippocampus no change was seen compared with controls. Neither site displayed differences at 3 months. This limited response could be due to sampling time and marker kinetics, as previous studies have seen robust increase in the first 24 hours following injury and a decreased to baseline by 48 hours [79]. An additional limiting factor may be the lack of severity of impact in our model, as the majority of brain injury literature examining TNF has done so in moderate or severe injuries using rodent brain or human CSF [80, 81].

CCL11 is a chemokine that is released by microglia and astrocytes as part of the inflammatory response following TBI [82]. Like other CNS inflammation mechanisms, CCL11 release is designed to assist with protection and repair, however increased levels of CCL11 is a driver of oxidative stress and excitotoxic pathways that precipitate synaptic dysfunction and neuronal death [83]. In previous mouse studies, increased CCL11 has been correlated with symptoms such as impaired cognition and memory [84], but to our knowledge this is the first study to assess CCL11 in a model of head acceleration mTBI. This is warranted as elevated levels of

CCL11 have been found in cortex of former American football players with CTE, with a correlation between CCL11 and tau pathology, and a significant association with the number of years playing football [85]. In a similar way, the present study found elevated CCL11 mRNA expression in the cortex at the chronic measurement time-point in mice that received 15 impacts, but not in the 1-IMP and 5-IMP groups. Expression was also significantly higher at 48 hours in the hippocampus, with level correlated with the number of head impacts received.

Considering the information provided by the mRNA expression in the brain, we sought to examine if serum protein could be detected that demonstrated similar damage signalling. In human blood biomarker mTBI research, the most promising measure of neurological damage is tau, and astrocyte activation is GFAP [86]. We measured the tau epitope phosphorylated at threonine 231, as this provides the most clinically relevant marker of human injury, which has been shown in animal models to be correlated with pathological symptoms and injury severity [87]. We failed to find a difference in p-tau between groups, and it may be that the injury severity in our model was insufficient to elicit change. It may also have been more appropriate to measure total tau in plasma, as this has been most commonly done in human studies [88–91]. This study also did not find a difference between groups in serum GFAP. There are challenges in the detection of CNS proteins in the blood related to low concentrations due to limited blood brain permeability, metabolic degradation and clearance, and contamination during sample preparation [92], and out findings were likely limited by these factors.

A limitation of this work is that protein levels of the genes of interest were not evaluated for the cortex and hippocampus. As the purpose of this study was to evaluate novel genes involved in known neurodegenerative pathways, protein-based pathology investigation was outside the scope of this work. As such, the preliminary findings described in this investigation should be expanded upon in further studies. Similarly, further details of the exact gene expression signalling mechanisms of inflammation and excitotoxicity (such as genes GRIA1 and CCL11) which are induced following repeated mTBI will need to be determined. It has been reported that the model used in this study is subject to greater variability in impact site than models using a stereotaxic device to secure the head [93]. This may result in variability in outcomes, however this compromise allows freedom of head movement that more closely reflects most human mTBI, and therefore gives more clinically relevant outcomes, and full discussion of this principle has been described [14]. Another limitation of the study is that only male mice were used, and that sex differences were not examined. NIH policy describes the importance of including both male and female animals in preclinical studies [94]. Previous animal model studies of mTBI have demonstrated diversity between sexes in outcomes including gene expression, development of pathological proteins, and behaviour [95]. The decision to include only males was dictated by economics in order to simplify the study design by reducing group numbers. We recognise that human mTBI is not male-centric, with studies showing the high rates of concussion sustained by female athletes [96], and that this was a missed opportunity to study sex-specific vulnerabilities.

In conclusion, this study provides further evidence in the role of repetitive subconcussive impacts in the vulnerability of the brain to injury and chronic neurodegeneration. This study used impact thresholds lower than previously reported in literature and confirmed behavioural detriment at acute testing that persisted to chronic impairment. Commonly assessed genetic markers were confirmed in this model, and inflammation and excitotoxic genes were implicated in the pathological cascade. In both behavioural and genetic data, there was evidence of a dose-dependent response where single impact had minimal effect, and the highest number of impacts resulted in neurological damage and decline. Nonetheless, there was no evidence of upregulation of serum proteins of disease at acute or chronic timepoints. Further

investigations are needed to examine systemic protein circulation, and the presence of histological evidence of disease.

This project was supported by the Australian Government Research Training Scheme. Funding to publish this study was provided by the Mackay Hospital and Health Service and the Mackay Institute of Research and Innovation. The funding sources was not involved in any of the following: study design; collection, analysis, and interpretation of data; writing of the report; or decision to submit the article for publication.

## Supporting information

**S1 Fig. A module map showing the connection among altered genes.** Edges have been defined based on co-expression, association in curated databases, or co-mentioned in publications.
(TIF)

## Acknowledgments

The authors thank Ms. Kristal Parmenter and Ms. Peta-Anne Clark, Central Queensland University, for assistance with behavioural testing and laboratory work.

## Author Contributions

**Conceptualization:** Matthew I. Hiskens.

**Data curation:** Matthew I. Hiskens.

**Formal analysis:** Matthew I. Hiskens.

**Investigation:** Matthew I. Hiskens.

**Methodology:** Matthew I. Hiskens, Rebecca K. Vella.

**Resources:** Andrew S. Fenning.

**Supervision:** Anthony G. Schneiders, Rebecca K. Vella, Andrew S. Fenning.

**Visualization:** Matthew I. Hiskens.

**Writing – original draft:** Matthew I. Hiskens.

**Writing – review & editing:** Matthew I. Hiskens, Rebecca K. Vella, Andrew S. Fenning.

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
