## [Decision Letter · Decision Letter 0]

12 Apr 2021

PONE-D-20-37245

Repetitive mild traumatic brain injury affects inflammation and excitotoxic mRNA expression at acute and chronic time-points

PLOS ONE

Dear Dr. Hiskens,

Thank you for submitting your manuscript to PLOS ONE. After careful consideration, we feel that it has merit but does not fully meet PLOS ONE’s publication criteria as it currently stands. Therefore, we invite you to submit a revised version of the manuscript that addresses the points raised during the review process.

We look forward to receiving your revised manuscript.

Kind regards,

Firas H Kobeissy, PhD

Academic Editor

PLOS ONE

Journal Requirements:

4. Please include your tables as part of your main manuscript and remove the individual files.

Please note that supplementary tables should be uploaded as separate "supporting information" files.

Additional Editor Comments:

Dear Dr. Hiskens,

the article was assessed by two experts in the area of TBI,. there is enthusiasm for this work and showed to carry scientific merit to he mTBI field, there are some minor comments pertaining to discussion and the selection of the genes. please check reviewer 1 and 2 for their comments

i would also suggest that an interaction map be constructed to show the connection among the altered genes.

Looking forwad,

FK

Reviewers' comments:

Reviewer's Responses to Questions

**Comments to the Author**

1. Is the manuscript technically sound, and do the data support the conclusions?

Reviewer #1: Yes

Reviewer #2: Partly

2. Has the statistical analysis been performed appropriately and rigorously? 

Reviewer #1: Yes

Reviewer #2: Yes

3. Have the authors made all data underlying the findings in their manuscript fully available?

Reviewer #1: Yes

Reviewer #2: Yes

4. Is the manuscript presented in an intelligible fashion and written in standard English?

Reviewer #1: Yes

Reviewer #2: Yes

5. Review Comments to the Author

Reviewer #1: In the present manuscript titled “Repetitive mild traumatic brain injury affects inflammation and excitotoxic MRNA expression at acute and chronic time-points” authors Dr. Hiskens et al., presented a case study work where role of repetitive subconcussive Impacts were assessed by mRNA analyses performed on brain tissue samples of mTBI (control; one, five and 15 impacts) mice, using quantitative RT-PCR. The main finding is, eight genes that were assessed (including the commonly used indicators of the head trauma; MAPT, GFAP, AIF1, TNF, and NEFL) showed expression changes indicative of excitotoxicity and inflammation genes, that were observed on location and follow-up duration. However, there was no difference in serum samples between the groups for tau and GFAP proteins. Additionally, behavioral changes were investigated to evaluate acute and chronic mTBI symptoms involving neurological function and spatial learning and memory. The use of mTBI model of mice that mimics the subtler/milder form of TBI (as opposed to obvious injury) is a big positive in this this study. The time points (48 hrs to 3 months) taken in to consideration also adds a lot of value to the data. This together can provide value data to similar researchers in the field.

The manuscript is written in a high quality style and the data generated supported the study hypothesis. The authors did a great job in communicating the results and the flow between each section is decent. The figures (including the schematic) was very helpful for the readers to understand the flow of the work. The statistical analysis was performed strategically (especially with such a large data set). Same with the behavioral data presentation. The authors listed limitations of the study which I found was very observant.

I recommend that this paper be accepted after minor questions below are answered.

1. The kits used for ptau and GFAP: Are these kits ‘validated’ for use with mice samples? I suspect that the high background may be suppressing the signals (especially with a milder injury model the changes may not be great). May be this is why there is no difference between mTBI VS control. Can I get a comment form the authors on this?

2. Why did not the authors look at other proteins in the serum (ex: Neurofilament light (NfL) protein)? The reason being that this protein is being looked at various forms of concussion and also over pressure exposure. Can I get a comment form the authors on this?

3. Did the author consider looking at CSF biochemical analysis for the proteins? Perhaps this may show some signals for the proteins of the study (as used in serum).

Reviewer #2: The manuscript describes a rodent model of repetitive sub-concussive injuries and its effect on molecular and behavioral changes at acute and chronic time points following the injury. The objective of this project is interesting and an animal model that can replicate the effect of repetitive sub-concussive injuries in humans is needed. However, there are several issues with the experimental design and interpretation of results.

1. What happens to the 25g weight at the impact. Is it controlled fall? If not, does the weight fall on the animal after the platform collapse? Is there any specific reason for using this model over the closed head CCI injury model which can be controlled relatively well with high reproducibility? How much the fall account for the head injury is not clear.

2. It appears that the animals were not perfused before the collection of the brain. If that is the case then the gene expression analysis can be confounded by the presence of blood cells especially TNF alpha.

3.Why did authors choose to evaluate these specific set of genes only. This is not clear.

4.Figure 3: The latency times for the 15 IMP group were quite high during impact 1-6 than the 1-5 impacts of the 5 IMP group. The possible cause of this difference is not clear because essentially both the groups were treated in the same way until the first 5 impacts yet there is a significant difference?

5. "15-IMP mice did not have performance differences compared with the 5-IMP group, and 1-IMP

and 5-IMP mice performance were not different to control" This statement is not clear. If 15 impact and 5 impacts did not have performance difference. If 5 IMP is not different from sham control, then how is 15 impacts different than sham control?

6. Was the PID 1 same for all the groups? In other words, did the authors conducted the NSS for all three groups together after the 15 impacts were over? if this is the case, then 1 IMP and the 5 IMP group would have a significant time for recovery compared to 15 IMP and it is likely that the difference in NSS is probably because of the test conducted immediately after the last injury and not necessarily because of the cumulative injury.

7. The results are not explained in section 3.5 for the results. The key findings should be explained for the ease of readers.

8. The lack of protein data reduced the significance of the molecular findings.

6. PLOS authors have the option to publish the peer review history of their article (what does this mean?). If published, this will include your full peer review and any attached files.

Reviewer #1: **Yes: **Bharani Thangavelu

Reviewer #2: No

---

## [Author Response · Author response to Decision Letter 0]

23 Apr 2021

Reviewer #1:

1) The kits used for ptau and GFAP: Are these kits ‘validated’ for use with mice samples? I suspect that the high background may be suppressing the signals (especially with a milder injury model the changes may not be great). May be this is why there is no difference between mTBI VS control. Can I get a comment form the authors on this?

The ptau and GFAP kits are validated for use in quantifying mouse proteins in ELISA as per the manufacturer’s instructions. We suspect the reviewer is correct that the combination of high background and mild injury has resulted in the negative finding.

2) Why did not the authors look at other proteins in the serum (ex: Neurofilament light (NfL) protein)? The reason being that this protein is being looked at various forms of concussion and also over pressure exposure. Can I get a comment form the authors on this?

We agree with the reviewer that the investigation of Nfl protein would have been a valuable addition in this study, as this protein is widely investigated in mild TBI. We were limited to investigating two proteins due to the volume of serum that was collected in each subject.

3) Did the author consider looking at CSF biochemical analysis for the proteins? Perhaps this may show some signals for the proteins of the study (as used in serum).

We agree with the reviewer that CSF protein analysis would have been a valuable measure of protein change following injury. We included serum levels as these measures will be used in human mTBI for diagnosis. Our laboratory does not currently have the technical expertise to sample CSF from a mouse, however future studies using CSF protein analysis will be worth pursuing in understanding the relationship of expression across these fluids.

Reviewer #2:

1. What happens to the 25g weight at the impact. Is it controlled fall? If not, does the weight fall on the animal after the platform collapse? Is there any specific reason for using this model over the closed head CCI injury model which can be controlled relatively well with high reproducibility? How much the fall account for the head injury is not clear.

The impact weight is tethered to the apparatus such that there is no unintended secondary impact can occur. This is explained in our methods section: “The impact weight was tethered to the guide tube by commercially available braided nylon line (Spear and Jackson, Melbourne, VIC, Australia), restricting the fall of the weight so that it could continue downward no more than 1 cm beyond the starting position of the dorsal surface of the skull, thereby avoiding unintentional secondary contact.”

A systematic review that we published in 2019 https://doi.org/10.1002/jnr.24472 identified this model previously described https://doi.org/10.1016/j.ajpath.2016.07.013 as ideal for mimicking the acceleration forces, impact speeds and biomechanical properties of human mild TBI while avoiding the need for surgical alteration to the head.

We certainly agree with Reviewer #2 that CCI is useful in single impact moderate-severe forms of TBI, however this method is not appropriate for repetitive TBI due to the opening and closing of the skull.

The fall following head impact is a short distance (10cm) onto a cushion, and the lab that first described this method ensure no additional injury https://doi.org/10.1016/j.ajpath.2016.07.013

2. It appears that the animals were not perfused before the collection of the brain. If that is the case then the gene expression analysis can be confounded by the presence of blood cells especially TNF alpha.

We agree that perfusion is often ideal before the collection of the brain. However, we consider that the most important factor in our molecular analysis is protecting the integrity of the RNA in tissue by undertaking rapid dissection and ensuring that the tissue is as cold as possible.

It is the perspective of our lab that because the perfusion procedure has a lag of a few minutes between cessation of effective circulation and cooling of the tissue, hypoxic and oxidative stress processes make endogenous RNases of the brain more active which could potentially result in transcriptional changes. For this reason, we do not perfuse before collecting tissue when the aim of the experiment is to analyse gene expression, instead we perform rapid dissection on an ice-cold surface with the tissue first rinsed and then bathed in artificial CSF.

Reviewer #2 makes a very good justification for removing blood in the vasculature which may introduce peripheral inflammatory mediators expressing biological markers similar to microglia. However, we prefer to accept the blood as the natural state of the tissue and control for this constant across all animals because of the variation in effectiveness of perfusion between different animals. We feel that a reasonable assumption is that tissue dissected without perfusion contains the same amount of blood between biological replicates, versus potential differences in tissue that is perfused.

An example of a recent publication from a different lab that uses the same technique of no perfusion in the mRNA expression of microglia is doi:10.1111/gbb.12736 

3.Why did authors choose to evaluate these specific set of genes only. This is not clear.

We thank Reviewer #2 for this point, there was a lack of clarity in our overview of decision to assess these genes. Although there are numerous genes we could have analysed, we selected specific genes for investigation based upon the role they may play in the specific secondary injury processes involved following mTBI, namely excitotoxicity, inflammation, and neurodegeneration.

The discussion has been reordered to allow for a more logical flow of information regarding genes. Additionally, the following information has been added to the discussion of the manuscript:

“GRIA1 was assessed as a measure of excitotoxicity, which is a trigger of neurodegenerative cascades and is implicated in the disruption of spatial working memory in the hippocampus. To examine axonal neurodegeneration, MAPT and NEFL were selected, as they are indicators of structural damage and are involved in chronic neurofibrillary tangle formation and neurofilament breakdown, respectively. TARDBP was selected for its implication in these neurodegenerative processes and role in protein signalling and organelle transport within the neuron. GFAP and AIF1 were used as a measure of activated astrocytes and microglia, respectively. Glial cells are activated in response to injury where they promote neuroinflammation that is aimed at neurological recovery and repair. Finally, TNF and CCL11 were used as measures of inflammation, with TNF a classic pro-inflammatory cytokine that has been measured in TBI conditions, and increased CCL11 production found in brain aging and disease.”

Additionally, a module map has been constructed to show the connection among the altered genes. This has been included as S1 fig.

4.Figure 3: The latency times for the 15 IMP group were quite high during impact 1-6 than the 1-5 impacts of the 5 IMP group. The possible cause of this difference is not clear because essentially both the groups were treated in the same way until the first 5 impacts yet there is a significant difference?

This is an astute observation by the reviewer, in line with the results that we reported on these groups “The 15-IMP group RR time was significantly greater than 5-IMP group following impacts 2 through 4, but not for impacts 1 and 5“. Our explanation of this centres on the schedule of the impacts for each group, which saw the 5-IMP group receive 2 days of impacts, followed by 2 days of no impacts, before the final 3 days of impacts. In contrast, the 15-IMP group received 3 days of impacts, followed by 2 days with no impacts, before receiving the next set of 3 impacts. In this way it seems that the earlier opportunity for non-impact days had protective effect on the latency times when impacts were recommenced. In the discussion we explained this difference: ‘While the 5-IMP group had RR times significantly higher than control, this group did not see the same peak as the 15-IMP group, suggesting that the additional time between impacts served as an opportunity for healing such that accumulated detriment was not seen.’ Additional clarification of this point is now made in the discussion.

5. "15-IMP mice did not have performance differences compared with the 5-IMP group, and 1-IMP and 5-IMP mice performance were not different to control" This statement is not clear. If 15 impact and 5 impacts did not have performance difference. If 5 IMP is not different from sham control, then how is 15 impacts different than sham control?

Thank you for picking up the lack of clarity in this statement. We believe these details provide minimal value to the presentation of results, and the sentence has been removed.

6. Was the PID 1 same for all the groups? In other words, did the authors conducted the NSS for all three groups together after the 15 impacts were over? if this is the case, then 1 IMP and the 5 IMP group would have a significant time for recovery compared to 15 IMP and it is likely that the difference in NSS is probably because of the test conducted immediately after the last injury and not necessarily because of the cumulative injury

Figure 2a shows the schedule of impacts for each of the groups across the 23 days of impact administration. The timing of the final impact for each group was the 23rd day of the study. The 1-IMP and 5-IMP groups were anesthetised but not impacted (designated on the figure by ‘A’) to match the timing of the impacts in the 15-IMP group. In this way, PID 1 was the first day after the final impact for every group – said another way, the NSS was administered 24 hrs after the final impact for every group.

7. The results are not explained in section 3.5 for the results. The key findings should be explained for the ease of readers.

The key findings have now been included in section 3.5:

In the acute cortex the 15-IMP group showed increased expression of MAPT, GFAP, and TNF genes relative to control (p >0.05). In the chronic cortex AIF1, CCL11 and TARDBP levels in the 15-IMP group were elevated relative to control (p >0.05). In the hippocampus, acute measurement saw the increase of GRIA1, CCL11, and NEFL in the 15-IMP group (p >0.05), while chronic measurement resulted in elevated GFAP levels in the 15-IMP group and elevated AIF1 in all impact groups relative to control (p >0.05).

8. The lack of protein data reduced the significance of the molecular findings.

We agree with the reviewer that the lack of protein data reduces the significance of the molecular findings. A statement of this limitation is present in the discussion: “A limitation of this work is that protein levels of the genes of interest were not evaluated for the cortex and hippocampus. As the purpose of this study was to evaluate novel genes involved in known neurodegenerative pathways, protein-based pathology investigation was outside the scope of this work. As such, the preliminary findings described in this investigation should be expanded upon in further studies.”

---

## [Editor Report · Decision Letter 1]

26 Apr 2021

Repetitive mild traumatic brain injury affects inflammation and excitotoxic mRNA expression at acute and chronic time-points

PONE-D-20-37245R1

Dear Dr. Hiskens,

We’re pleased to inform you that your manuscript has been judged scientifically suitable for publication and will be formally accepted for publication once it meets all outstanding technical requirements.

Kind regards,

Firas H Kobeissy, PhD

Academic Editor

PLOS ONE
---

## [Editor Report · Acceptance letter]

28 Apr 2021

PONE-D-20-37245R1 

Repetitive mild traumatic brain injury affects inflammation and excitotoxic mRNA expression at acute and chronic time-points 

Dear Dr. Hiskens:

I'm pleased to inform you that your manuscript has been deemed suitable for publication in PLOS ONE. Congratulations! Your manuscript is now with our production department. 

Kind regards, 

on behalf of

Dr. Firas H Kobeissy 

Academic Editor

PLOS ONE